# Learning to Remember Patterns: Pattern Matching Memory Networks for Traffic Forecasting

**Hyunwook Lee, Seungmin Jin, Hyeshin Chu, Hongkyu Lim, and Sungahn Ko**[*]
Ulsan National Institute of Science and Technology
{gusdnr0916,skyjin,hyeshinchu,limhongkyu1219,sako}@unist.ac.kr

## Abstract

Traffic forecasting is a challenging problem due to complex road networks and sudden speed changes caused by various events on roads. Several models have been proposed to solve this challenging problem, with a focus on learning the spatio-temporal dependencies of roads. In this work, we propose a new perspective for converting the forecasting problem into a pattern-matching task, assuming that large traffic data can be represented by a set of patterns. To evaluate the validity of this new perspective, we design a novel traffic forecasting model called Pattern-Matching Memory Networks (PM-MemNet), which learns to match input data to representative patterns with a key-value memory structure. We first extract and cluster representative traffic patterns that serve as keys in the memory. Then, by matching the extracted keys and inputs, PM-MemNet acquires the necessary information on existing traffic patterns from the memory and uses it for forecasting. To model the spatio-temporal correlation of traffic, we proposed a novel memory architecture, GCMem, which integrates attention and graph convolution. The experimental results indicate that PM-MemNet is more accurate than state-of-the-art models, such as Graph WaveNet, with higher responsiveness. We also present a qualitative analysis describing how PM-MemNet works and achieves higher accuracy when road speed changes rapidly.

## 1 Introduction

Traffic forecasting is a challenging problem due to complex road networks, varying patterns in the data, and intertwined dependencies among models. This implies that prediction methods should not only find intrinsic spatio-temporal dependencies among many roads, but also quickly respond to irregular congestion and various traffic patterns (Lee et al., 2020) caused by external factors, such as accidents or weather conditions (Vlahogianni et al., 2014; Li & Shahabi, 2018; Xie et al., 2020; Jiang & Luo, 2021). To resolve these challenges and successfully predict traffic conditions, many deep learning models have been proposed. Examples include the models with graph convolutional neural networks (GCNs) (Bruna et al., 2014) and recurrent neural networks (RNNs) (Siegelmann & Sontag, 1991), which outperform conventional statistical methods such as autoregressive integrated moving average (ARIMA) (Vlahogianni et al., 2014; Li et al., 2018). Attention-based models, such as GMAN (Zheng et al., 2020), have also been explored to better handle complex spatio-temporal dependency of traffic data. Graph WaveNet (Wu et al., 2019) adopts a diffusion process with a self-learning adjacency matrix and dilated convolutional neural networks (CNNs), achieving state-of-the-art performance. Although effective, existing models have a weakness in that they do not accurately forecast when conditions are abruptly changed (e.g., rush hours and accidents).

In this work, we aim to design a novel method for modeling the spatio-temporal dependencies of roads and to improve forecasting performance. To achieve this goal, we first extract representative traffic patterns from historical traffic data, as we find that there are similar traffic patterns among roads, and a set of traffic patterns can be generalized for roads with similar spatio-temporal features. Figure 1 shows the example speed patterns (left, 90-minute window) that we extract from many

---

[*]Corresponding Author

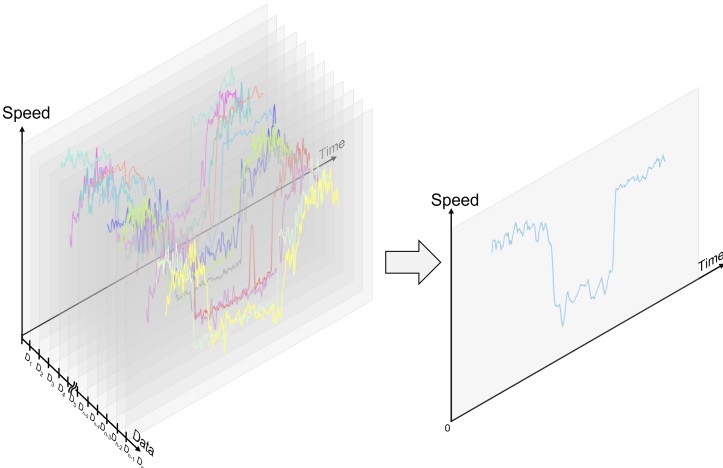

Figure 1: (left) Multiple traffic data with similar pattern, (right) extracted representative pattern

different roads and a representative traffic pattern (right time series). With the representative patterns, we transform the conventional forecasting problem into a pattern-matching task to find out which pattern would be the best match for the given spatio-temporal features to predict future traffic conditions. With insights from the huge success of neural memory networks in natural language processing and machine translation (Weston et al., 2015; Sukhbaatar et al., 2015; Kaiser et al., 2017; Madotto et al., 2018), we design graph convolutional memory networks called GCMem to manage representative patterns in spatio-temporal perspective. Lastly, we design PM-MemNet, which utilizes representative patterns from GCMem for traffic forecasting. PM-MemNet consists of an encoder and a decoder. The encoder consists of temporal embedding with stacked GCMem, which generates meaningful representations via memorization, and the decoder is composed of a gated recurrent unit (GRU) with GCMem. We compare PM-MemNet to existing state-of-the-art models and find that PM-MemNet outperforms existing models. We also present a qualitative analysis in which we further investigate the strengths of PM-MemNet in managing a traffic pattern where high responsiveness of a model to abrupt speed changes is desired for accurate forecasting.

The experimental results indicate that PM-MemNet achieves state-of-the-art performance, especially in long-term prediction, compared to existing deep learning models. To further investigate the characteristics of PM-MemNet, we conduct an ablation study with various decoder architectures and find that PM-MemNet demonstrates the best performance. We also investigate how the number of representative patterns affects model performance. Finally, we discuss the limitations of this work and future directions for neural memory networks in the traffic forecasting domain.

The contributions of this work include: (1) computing representative traffic patterns of roads, (2) design of GCMem to manage the representative patterns, (3) presenting a novel traffic prediction model, PM-MemNet, that matches and uses the most appropriate patterns from GCMem for traffic forecasting, (4) evaluation of PM-MemNet compared to state-of-the-art models, (5) qualitative analysis to identify the strengths of PM-MemNet, and (6) discussion of limitations and future research directions.

## 2 RELATED WORK

### 2.1 TRAFFIC FORECASTING

Deep learning models achieve huge success by effectively capturing spatio-temporal features in traffic forecasting tasks. Past studies ahve shown that RNN-based models outperform conventional temporal modeling approaches, such as ARIMA and support vector regression (SVR) (Vlahogianni et al., 2014; Li et al., 2018). More recently, many studies have demonstrated that attention-based models (Zheng et al., 2020; Park et al., 2020) and CNNs (Yu et al., 2018; Wu et al., 2019) record

better performance in long-term period prediction tasks, compared to RNN-based models. In terms of spatial modeling, Zhang et al. (2016) propose a CNN-based spatial modeling method for Euclidean space. Another line of modeling methods, such as GCNs, using graph structures for managing complex road networks also become popular. However, there are difficulties in using GCNs in the modeling process, such as the need to build an adjacency matrix and the dependence of GCNs on invariant connectivity in the adjacency matrix. To overcome these difficulties, a set of approaches, such as graph attention models (GATs), have been proposed to dynamically calculate edge importance (Park et al., 2020). GWNet (Wu et al., 2019) adopts a self-adaptive adjacency matrix to capture hidden spatial dependencies in training. Although effective, forecasting models still suffer from inaccurate predictions due to abruptly changing road speeds and instability, with lagging patterns in long-term periods. To address these challenges, we build, save, and retrieve representative traffic patterns for predicting speed rather than directly forecasting with an input sequence.

## 2.2 Neural Memory Networks

Neural memory networks are widely used for sequence-to-sequence modeling in the natural language processing and machine translation domains. Memory networks are first proposed by Weston et al. (2015) to answer a query more precisely even for large datasets with long-term memory. Memory networks perform read and write operations for given input queries. Sukhbaatar et al. (2015) introduce end-to-end memory networks that can update memory in an end-to-end manner. Through the end-to-end memory learning, models can be easily applied to realistic settings. Furthermore, by using adjacent weight tying, they can achieve recurrent characteristics that can enhance generalization. Kaiser et al. (2017) propose novel memory networks that can be utilized in various domains where life-long one-shot learning is needed. Madotto et al. (2018) also introduce Mem2Seq, which integrates the multi-hop attention mechanism with memory networks. In our work, we utilize memory networks for traffic pattern modeling due to the similarity of the tasks and develop novel graph convolutional memory networks called GCMem to better model the spatio-temporal correlation of the given traffic patterns.

## 3 Proposed Approach

In this section, we define the traffic forecasting problem, describe how we extract key patterns in the traffic data that serve as keys, and introduce our model, PM-MemNet.

### 3.1 Problem Setting

To handle the spatial relationships of roads, we utilize a road network graph. We define a road network graph as $\mathcal{G} = (\mathcal{V}, \mathcal{E}, \mathcal{A})$, where $\mathcal{V}$ is a set of all different nodes with $|\mathcal{V}| = N$, $\mathcal{E}$ is a set of edges representing the connectivity between nodes, and $\mathcal{A} \in \mathbb{R}^{N \times N}$ is a weighted adjacency matrix that contains the connectivity and edge weight information. An edge weight is calculated based on the distance and direction of the edge between two connected nodes. As used in the previous approaches (Li et al., 2018; Wu et al., 2019; Zheng et al., 2020; Park et al., 2020), we calculate edge weights via the Gaussian kernel as follows: $A_{i,j} = \exp\left(-\frac{\text{dist}_{ij}^2}{\sigma^2}\right)$, where $\text{dist}_{ij}$ is the distance between node $i$ and node $j$ and $\sigma$ is the standard deviation of the distances.

Prior research has formulated a traffic forecasting problem as a simple spatio-temporal data prediction problem (Li et al., 2018; Wu et al., 2019; Zheng et al., 2020; Park et al., 2020) aiming to predict values in the next $T$ time steps using previous $T'$ historical traffic data and an adjacency matrix. Traffic data at time $t$ is represented by a graph signal matrix, $X_{\mathcal{G}}^t \in \mathbb{R}^{N \times d_{in}}$, where $d_{in}$ is the number of features, such as speed, flow, and time of the day. In summary, the goal of the previous work is to learn a mapping function $f(\cdot)$ to directly predict future $T$ graph signals from $T'$ historical input graph signals:

$$\left[X_{\mathcal{G}}^{(t-T'+1)}, \ldots, X_{\mathcal{G}}^{(t)}\right] \xrightarrow{f(\cdot)} \left[X_{\mathcal{G}}^{(t+1)}, \ldots, X_{\mathcal{G}}^{(t+T)}\right]$$

The goal of this study is different from previous work in that we aim to predict future traffic speeds from patterned data, instead of utilizing input $X_{\mathcal{G}}$ directly. We denote $\mathbb{P} \subset \mathbb{R}^{T'}$ as a set of representative traffic patterns, $p \in \mathbb{P}$ as one traffic pattern in $\mathbb{P}$, and $d : \mathbb{X} \times \mathbb{P} \to [0, \infty)$ as a distance

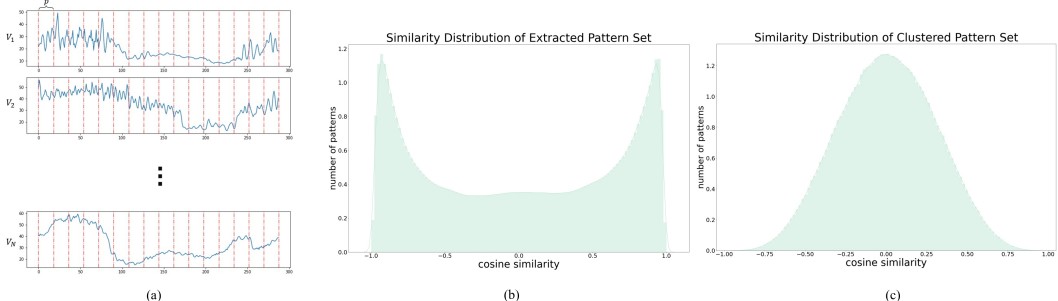

Figure 2: (a) Example daily patterns. Each part between the red dash lines is the speed patterns sliced by a given time window, (b) cosine similarity distribution of the original pattern set with class imbalance, and (c) the clustered pattern set.

function for pattern matching. Detailed information about traffic pattern extraction will be discussed in the next subsection. Our problem is to train the mapping function $f(\cdot)$ as follows:

$$\left[X_{\mathcal{G}}^{(t-T'+1)}, \ldots, X_{\mathcal{G}}^{(t)}\right] \xrightarrow{d(\cdot), k-NN} \left[P_1^t, \ldots, P_N^t\right] \xrightarrow{f(\cdot)} \left[X_{\mathcal{G}}^{(t+1)}, \ldots, X_{\mathcal{G}}^{(t+T)}\right],$$

where $P_i^t = \{p_1, \ldots, p_k\}$ is a set of $k$-nearest neighboring traffic patterns of node $i$ in time $t$, with a distance function $d$. Note that $p_j$ is the $j$-th nearest neighbor pattern.

## 3.2 KEY EXTRACTION FROM TRAFFIC PATTERNS

Analyzing the traffic data, we find that the data has repeating patterns. In traffic data, the leading and trailing patterns have a high correlation, even during short-term periods. To take advantage of these findings, in our model, we build a representative pattern set, $\mathbb{P}$. First, from historical data, we compute an average daily pattern, which consists of 288 speed data points (total 24 hours with 5-minute intervals) for each vertex $v \in \mathcal{V}$. We then extract pattern $p$ by slicing the daily patterns with a given window size $T'$, as shown in Figure 2 (a). At this stage, $|\mathbb{P}| = N \times \lfloor \frac{288}{T'} \rfloor$. After we collect the patterns, we investigate similarity distribution of the extracted pattern set, $\mathbb{P}$, via cosine similarity (Figure 2 (b)) and find that the pattern set $\mathbb{P}$ has a biased distribution with too many similar patterns (i.e., class imbalance). Since such class imbalance causes memory ineffectiveness in accurate memory retrieval and gives biased training results, we use clustering-based undersampling (Lin et al., 2017) with cosine similarity. For example, if pattern $p$ and pattern $p'$ have a cosine similarity larger than $\delta$, they are in same cluster. We utilize the center of each cluster as a representative pattern of that cluster. After undersampling by clustering, we have a balanced and representative pattern set, $\mathbb{P}$, as shown in Figure 2 (c), which we use as keys for memory access. Table 2 presents the effect of different $\delta$ and $|\mathbb{P}|$ on forecasting performance.

## 3.3 NEURAL MEMORY ARCHITECTURE

Conventionally, memory networks have used the attention mechanism for memory units to enhance memory reference performance, but this attention-only approach cannot effectively capture spatial dependencies among roads. To address this issue, we design a new memory architecture, GCMem (Figure 3 (b)), which integrates multi-layer memory with the attention mechanism (Madotto et al., 2018) and graph convolution (Bruna et al., 2014). By using GCMem, a model can capture both pattern-level attention and graph-aware information sharing via GCNs.

To effectively handle representative patterns from a spatio-temporal perspective, we utilize several techniques. First, we use a modified version of the adjacent weight tying technique in MemNN (Sukhbaatar et al., 2015; Madotto et al., 2018), which has been widely used for sentence memorization and connection searches between query and memorized sentences. Sukhbaatar et al. (2015); Madotto et al. (2018) propose the technique to capture information from memorized sentences by making use of sentence-level attention. However, their methodology only learns pattern similarity and cannot handle spatial dependency. Using the same method in traffic forecasting is insufficient since handling a graph structure is essential for building spatial dependencies of roads. In

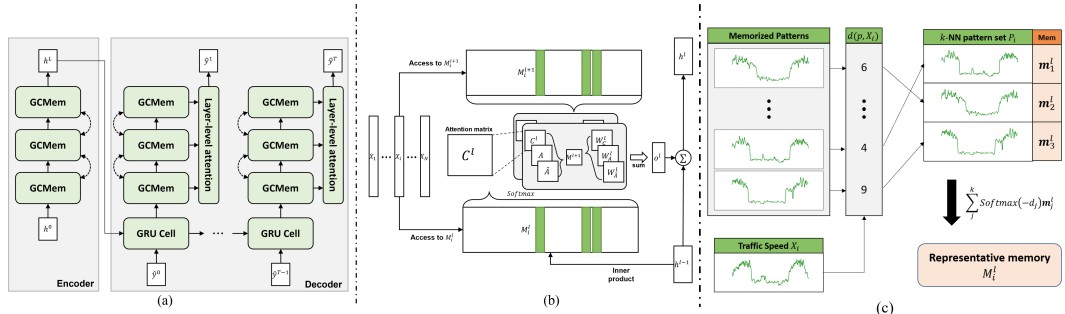

Figure 3: (a) Overall architecture of PM-MemNet where $L = 3$. Dashed line means adjacent weight tying. (b) GCMem architecture with GCNs (gray blocks) and (c) representative memory selection among $k$-nearest patterns for $X_i$ with $k$=3, $d_j = d(p_j, X_i)$.

order to consider the graph structure while maintaining the original sentence-level attention score, we use an adjacency matrix, a learnable adaptive matrix, and attention scores for the GCNs. By maintaining pattern-level attention, the model takes advantage of both pattern-level information sharing and adjacent weight tying (Madotto et al., 2018). As a result, adjacent memory cells can effectively retain attention mechanisms while considering a graph structure.

Figure 3 (a) and (b) shows PM-MemNet and our proposed graph convolution memory architecture. The memories for the GCMem are embedding matrices $\mathbf{M} = \{\mathbf{M}^1, \dots, \mathbf{M}^{L+1}\}$, where $\mathbf{M}^l \in \mathbb{R}^{|\mathbb{P}| \times d_h}$. For memory reference, we utilize pattern set $\mathbb{P}$, which contains the extracted traffic patterns (in Section 3.2) and $k$-Nearest Neighbor ($k$-NN) with distance function, $d(\cdot)$. For each input traffic data $X_i \in \mathbb{R}^{T' \times d_{in}}$ in node $i$, based on $k$-nearest patterns and the distance function, we build a representative memory $M_i^l$ as follows:

$$M_i^l = \sum_{j=1}^{k} \text{Softmax}(-d_j)\mathbf{m}_j^l, \tag{1}$$

where $\mathbf{m}_j^l = \mathbf{M}^l(p_j)$ is the memory context for $p_j$ and layer $l$, $d_j = g(X_i, p_j)$, and $\text{Softmax}(z_i) = e^{z_i} / \sum_j e^{z_j}$. We summarize our representative memory selection process in Figure 3 (c).

After calculating representative memory, GCMem calculates hidden state $h^l \in \mathbb{R}^{N \times d_h}$ with previous hidden state $h^{l-1}$ and representative memory $M^l$ and $M^{l+1}$ (Figure 3 (b)). For each representative memory $M_j^l$, PM-MemNet calculates pattern-level attention scores, $\alpha_{i,j}^l$ as follows:

$$\alpha_{i,j}^l = \text{Softmax}\left(\frac{h_i^{l-1}(M_j^l)^\top}{\sqrt{d_h}}\right), \tag{2}$$

where $h_i^{l-1} \in \mathbb{R}^{d_h}$ is the previous hidden state of node $i$. We denote an attention matrix as $C^l \in \mathbb{R}^{N \times N}$, where $\alpha_{i,j}^l$ is the entry in the $i$-th row and $j$-th column of $C^l$. Then, given memory unit $M^{l+1} \in \mathbb{R}^{N \times d_h}$ for the next layer $l + 1$, we calculate output feature, $o^l$ with graph convolution as shown below:

$$o^l = \sum_i^h \left(W_{A,i}^l M^{l+1} A + W_{\tilde{A},i}^l M^{l+1} \tilde{A} + W_{C,i}^l M^{l+1} C^l\right), \tag{3}$$

where $A \in \mathbb{R}^{N \times N}$ is an adjacency matrix and $\tilde{A} = \text{Softmax}(ReLU(E_1 E_2^\top))$ is a learnable matrix, which captures hidden spatio-temporal connections (Wu et al., 2019; Shi et al., 2019). $E_1, E_2 \in \mathbb{R}^{N \times d_h}$ are learnable node embedding vectors and $W \in \mathbb{R}^{d_h}$ is a learnable matrix. We use $ReLU$ as an activation function. Then, we update hidden states by $h^l = h^{l-1} + o^l$. Before we update the hidden states, we apply batch normalization on $o^l$.

## 3.4 ENCODER ARCHITECTURE

The left side of Figure 3 (a) shows the proposed encoder architecture. PM-MemNet handles traffic patterns and their corresponding memories. Although the patterns in the memory provide enough

information for training, there are other types of data that can also be used for prediction. For example, different roads have their own patterns that may not be captured in advance (e.g., the unique periodicity of roads around an industry complex (Lee et al., 2020)). In addition, there would be some noise and anomalous patterns due to various events (e.g., accidents), which are not encoded when patterns are grouped. As such, we provide embedding for the time (i.e., $emb$) and noise (i.e., $N_i$) that the encoder uses to generate input query $h^0$ for GCMem. Specifically, for the time series $\mathbf{T} = [t - T' + 1, \ldots, t]$ and noise $N_i = X_i - p_1$, we calculate its representation, $h_i^0$ as shown below:

$$h_i^0 = emb(T) + W_n N_i, \qquad (4)$$

where $emb$ and $W_n$ represent a learnable embedding for the time of day and a learnable matrix, respectively. Note that $emb(T) \in \mathbb{R}^{d_h}$ and $W_n \in \mathbb{R}^{d_h \times T}$ PM-MemNet updates $h_i^0$ using $L$-layer GCMem. We use the output of the encoder $h^L \in \mathbb{R}^{N \times d_h}$ as an initial hidden state in the decoder.

## 3.5 DECODER ARCHITECTURE

As shown in Figure 3 (a), we build our decoder architecture using single layer GRU, followed by the $L$-stacked GCMem. For each $t$ step, the decoder predicts the value at time step $t$ using the previous prediction $\hat{y}^{t-1}$ and GRU hidden state $\tilde{h}_{t-1}$. Initially, $\hat{y}^0$ is zero matrix and $\tilde{h}_0$ is encoder hidden state $h^L$. The hidden states from the GRU will be an input for the $L$-stacked GCMem. Similar to our encoder architecture, GCMem updates hidden states with attention and GCNs. Instead of using updated hidden states for prediction directly, we utilize layer-level self-attention in the decoder. Specifically, for each GCMem layer $l$ and node $i$, we calculate energy, $e_{i,l}$ using the previous hidden state $h_i^{l-1}$ and memory context $M_i^l$ as shown below:

$$e_{i,l} = \frac{(h_i^{l-1})(M_i^l W^l)^\top}{\sqrt{d_h}}, \qquad (5)$$

where $d_h$ is the hidden size and $W^l \in mathbbR^{d_h \times d_h}$ is learnable matrix. Then, with the output feature $o_i^l$ of each layer $l$ and node $i$, we can predict $\hat{y}_i^t$ as:

$$\hat{y}_i^t = \sum_l^L \alpha_{i,l} o_i^l W_{proj}, \qquad (6)$$

where $W_{proj} \in \mathbb{R}^{d_h \times d_{out}}$ is a projection layer and $\alpha_{i,l} = \text{Softmax}(e_{i,l})$. Note that $h^0$ is equivalent to a hidden state of a GRU cell, $\tilde{h}_t$. Using layer-level attention, PM-MemNet utilizes information from each layer more effectively.

## 4 EVALUATION

In this section, we explain the experiments conducted to compare PM-MemNet to existing models in terms of accuracy. We use two datasets for the experiment–METR-LA and NAVER-Seoul[2]. METR-LA contains 4-month speed data from 207 sensors of Los Angeles highways (Li et al., 2018). NAVER-Seoul has 3-month speed data collected from 774 links in Seoul, Korea. As NAVER-Seoul data covered the main arterial roads in Seoul, it can be considered a more difficult dataset with many abruptly changing speed patterns compared to METR-LA data. Both datasets have five-minute interval speeds and timestamps. Before training the PM-MemNet, we have filled out missing values using historical data and applied z-score normalization. We use 70% of the data for training, 10% for validation, and the rest for evaluation, as Li et al. (2018); Wu et al. (2019); Zheng et al. (2020) have done in their work.

## 4.1 EXPERIMENTAL SETUP

In our experiment, we use 18 sequence data points as a model input ($T' = 18$, one and a half hours) and predict the next 18 sequences. For the $k$-NN, we utilize the cosine similarity function for the similarity measurement and set $k = 3$. To extract $\mathbb{P}$, we utilize the training dataset and initialize parameters and embedding using Xavier initialization. After performing a greedy search among $d_h = [16, 32, 64, 128]$, $L = [1, 2, 3, 4]$, and various $|\mathbb{P}|$ values, we set $d_h$ as 128, $L$ as 3, and

Table 1: Experimental Results for NAVER-Seoul and METR-LA datasets

| Dataset | T | Metric | HA | MLP | STGCN | GCRNN | DCRNN | ASTGCN | GWNet | GMAN | PM-MemNet |
|---|---|---|---|---|---|---|---|---|---|---|---|
| NAVER-Seoul | 15min | MAE | 6.54 | 5.28 | 4.63 | 4.87 | 4.86 | 5.09 | 4.91 | 5.20 | **4.57** |
| | | MAPE | 18.24 | 16.86 | 14.49 | 15.23 | 15.35 | 16.14 | 14.86 | 16.98 | **14.43** |
| | | RMSE | 9.32 | 7.78 | 6.92 | 7.18 | 7.12 | 7.44 | 7.24 | 8.32 | **6.72** |
| | 30min | MAE | 7.16 | 6.13 | 5.50 | 5.73 | 5.67 | 5.71 | 5.26 | 5.35 | **5.04** |
| | | MAPE | 20.15 | 20.05 | 17.37 | 18.17 | 18.38 | 18.78 | **16.16** | 17.47 | 16.34 |
| | | RMSE | 10.18 | 9.51 | 8.83 | 9.03 | 8.80 | 8.73 | 8.13 | 8.67 | **7.86** |
| | 60min | MAE | 8.22 | 7.08 | 6.77 | 6.58 | 6.40 | 6.22 | 5.55 | 5.48 | **5.24** |
| | | MAPE | 23.37 | 23.44 | 20.42 | 20.95 | 21.09 | 20.37 | 16.97 | 17.89 | **16.94** |
| | | RMSE | 11.54 | 11.13 | 10.89 | 10.58 | 10.06 | 9.58 | 8.77 | 8.94 | **8.39** |
| | 90min | MAE | 9.24 | 7.79 | 8.06 | 7.14 | 6.86 | 6.76 | 5.87 | 5.58 | **5.40** |
| | | MAPE | 26.40 | 26.08 | 22.93 | 22.86 | 22.74 | 21.83 | 17.89 | 18.18 | **17.44** |
| | | RMSE | 12.77 | 12.17 | 12.86 | 11.43 | 10.69 | 10.32 | 9.33 | 9.09 | **8.68** |
| METR-LA | 15min | MAE | 4.23 | 2.93 | 2.61 | 2.59 | **2.56** | 3.25 | 2.72 | 2.86 | 2.66 |
| | | MAPE | 9.76 | 7.76 | **6.59** | 6.73 | 6.67 | 9.27 | 7.14 | 7.67 | 7.06 |
| | | RMSE | 7.46 | 5.81 | 5.19 | **5.12** | 5.10 | 6.28 | 5.20 | 5.77 | 5.28 |
| | 30min | MAE | 4.80 | 3.60 | 3.22 | 3.08 | **3.01** | 3.80 | 3.12 | 3.14 | 3.02 |
| | | MAPE | 11.30 | 10.00 | **8.39** | 8.72 | 8.42 | 11.28 | 8.66 | 8.79 | 8.49 |
| | | RMSE | 8.34 | 7.29 | 6.63 | 6.32 | 6.29 | 7.59 | 6.34 | 6.54 | **6.28** |
| | 60min | MAE | 5.80 | 4.69 | 4.31 | 3.74 | 3.60 | 4.49 | 3.58 | 3.48 | **3.40** |
| | | MAPE | 14.04 | 13.68 | 11.13 | 11.50 | 10.73 | 13.69 | 10.30 | 10.10 | **9.88** |
| | | RMSE | 9.86 | 9.24 | 8.71 | 7.71 | 7.65 | 8.94 | 7.53 | 7.30 | **7.24** |
| | 90min | MAE | 6.65 | 5.58 | 5.41 | 4.23 | 4.06 | 4.97 | 3.85 | 3.71 | **3.64** |
| | | MAPE | 16.37 | 17.08 | 13.76 | 13.49 | 12.53 | 15.53 | 11.39 | 11.00 | **10.74** |
| | | RMSE | 10.97 | 10.52 | 10.47 | 8.79 | 8.58 | 9.71 | 8.12 | 7.71 | **7.74** |

$|\mathbb{P}| \approx 100$ with $\delta = 0.7$ and $0.9$ for NAVER-Seoul and METR-LA, respectively. Table 2 presents the experimental results with different memory and pattern sizes (i.e., $|\mathbb{P}|$) We apply the Adam optimizer with a learning rate of 0.001 and use the mean absolute error (MAE) as a loss function.

We compare PM-MemNet to the following baseline models: (1) multilayer perceptron (MLP); (2) STGCN (Yu et al., 2018), which forecasts one future step using graph convolution and CNNs; (3) Graph Convolution Recurrent Neural Network (GCRNN); (4) DCRNN (Li et al., 2018), a sequence-to-sequence model that combines diffusion convolution in the gated recurrent unit; (5) AST-GCN (Guo et al., 2019), which integrates GCN, CNN, and spatial and temporal attention; (6) Graph-WaveNet (GWNet) (Wu et al., 2019), which forecasts multiple steps at once by integrating graph convolution and dilated convolution; and (7) Graph Multi-Attention Network (GMAN) (Zheng et al., 2020) which integrates spatial and temporal attention with gated fusion mechanism. GMAN also predicts multiple steps at once. To allow detailed comparisons, we train the baseline models to forecast the next 90 minutes of speeds at 5-minute intervals, given the past 18 5-minute interval speed data points. We train the baseline models in an equivalent environment with PyTorch[1] using the public source codes and settings provided by the authors. Detailed settings, including the hyperparameters, are available in Appendix A.1. To ease the performance comparison with previous work, we provide additional experimental results with a commonly used setting, where $T' = T = 12$, in Table 6 in Appendix. We also present a qualitative analysis of our method for NAVER-Seoul dataset in Appendix A.4 Our source codes are available on GitHub[2].

## 4.2 EXPERIMENTAL RESULTS

Table 1 displays the experimental results for NAVER-Seoul and METR-LA for the next 15 minutes, 30 minutes, 60 minutes, and 90 minutes using mean absolute error (MAE), mean absolute percentage error (MAPE), and root mean square error (RMSE). Note that Table 3 and Figure 5 in Appendix show more detailed experimental results and error information. As Table 1 shows, PM-MemNet achieves state-of-the-art performance for both datasets. Specifically, PM-MemNet yields the best performance in all intervals with NAVER-Seoul data, while it outperforms other models in the long-term prediction (i.e., 60 and 90 minutes) with METR-LA data.

In both datasets, we find interesting observations. First, RNN-based models perform better than other models for short term periods (i.e., 15 minutes), but they show a weakness for long-term periods. This problem occurs in sequence-to-sequence RNNs due to error accumulation caused by its auto-regressive property. Compared to the RNN-based models, PM-MemNet shows a lesser perfor-

---

[1] https://pytorch.org
[2] https://github.com/HyunWookL/PM-MemNet

Table 2: Ablation study result. Note that 'Ours' means PM-MemNet.

| Dataset | T | Metric | Ours | SimpleMem | CNN Decoder | RNN Decoder | Ours (L=1) | Ours ($\mathbb{P} = k$) | Ours ($\mathbb{P} >> 1000$) |
|---|---|---|---|---|---|---|---|---|---|
| NAVER-Seoul | 15min | MAE | 4.57 | 5.72 | **4.56** | 4.67 | 4.72 | 4.66 | 4.59 |
| | | MAPE | 14.43 | 18.18 | **14.40** | 14.84 | 14.98 | 14.77 | 14.48 |
| | | RMSE | 6.72 | 8.79 | **6.71** | 6.83 | 6.87 | 6.89 | 6.73 |
| | 30min | MAE | **5.04** | 5.84 | 5.06 | 5.19 | 5.22 | 5.21 | 5.09 |
| | | MAPE | **16.34** | 18.86 | 16.36 | 16.87 | 16.97 | 16.91 | 16.41 |
| | | RMSE | **7.86** | 9.24 | 7.90 | 8.02 | 8.03 | 8.18 | 7.97 |
| | 60min | MAE | **5.24** | 6.38 | 5.32 | 5.47 | 5.52 | 5.53 | 5.36 |
| | | MAPE | **16.94** | 21.42 | 17.19 | 17.91 | 17.97 | 18.05 | 17.19 |
| | | RMSE | **8.39** | 10.08 | 8.51 | 8.69 | 8.70 | 8.92 | 8.67 |
| | 90min | MAE | **5.40** | 6.95 | 5.55 | 5.70 | 5.72 | 5.74 | 5.52 |
| | | MAPE | **17.44** | 23.89 | 17.99 | 18.73 | 18.63 | 18.73 | 17.76 |
| | | RMSE | **8.68** | 10.88 | 8.82 | 9.10 | 9.05 | 9.31 | 8.94 |
| METR-LA | 15min | MAE | 2.66 | 3.01 | **2.63** | 2.68 | 2.68 | 2.67 | 2.68 |
| | | MAPE | 7.06 | 8.03 | **6.98** | 7.10 | 7.11 | 7.09 | 7.13 |
| | | RMSE | **5.28** | 5.94 | 5.32 | 5.31 | 5.31 | 5.35 | 5.31 |
| | 30min | MAE | 3.02 | 3.27 | **3.01** | 3.06 | 3.06 | 3.06 | 3.04 |
| | | MAPE | 8.49 | 9.20 | **8.46** | 8.56 | 8.59 | 8.67 | 8.51 |
| | | RMSE | 6.28 | 6.68 | 6.36 | 6.32 | **6.27** | 6.36 | 6.32 |
| | 60min | MAE | **3.40** | 3.72 | 3.41 | 3.46 | 3.47 | 3.49 | 3.45 |
| | | MAPE | **9.88** | 10.94 | **9.88** | 10.02 | 10.07 | 10.34 | 9.87 |
| | | RMSE | **7.24** | 7.70 | 7.28 | 7.31 | 7.25 | 7.39 | 7.32 |
| | 90min | MAE | **3.64** | 4.09 | 3.65 | 3.71 | 3.73 | 3.75 | 3.69 |
| | | MAPE | **10.74** | 12.25 | 10.85 | 10.87 | 10.98 | 11.30 | 10.63 |
| | | RMSE | 7.74 | 8.38 | **7.71** | 7.81 | 7.75 | 7.91 | 7.81 |

mance decrease, although it had decoder based on RNN architecture. This is because of GCMem and pattern data, which make representations for traffic forecasting more robust than other methods. In the case of NAVER-Seoul, we observe that all models suffer from decreased accuracy due to more complicated urban traffic patterns and road networks than those in METR-LA. In spite of these difficulties, PM-MemNet proves its efficiency with representative traffic patterns and the memorization technique. This result strengthens our hypothesis that the traffic data used for traffic forecasting can be generalized with a small number of pattern even when the traffic data are complex and hard to forecast.

## 4.3 ABLATION STUDY

We further evaluate our approach by conducting ablation studies (Table 2, Table 5 in Appendix). First, we check whether GCMem can effectively model spatio-temporal relationships among road networks. To this end, we compare the performance of PM-MemNet to that of SimpleMem, a simplified version of PM-MemNet in which the memory layer depends only on pattern-level attention and does not consider any graph-based relationship. By comparing PM-MemNet and SimpleMem, we observe that SimpleMem shows nearly 10% decrease in performance, which shows the importance of a graph structure in modeling traffic data. In the same context, according to Table 1 and Table 2, Simple-Mem has lower accuracy than previous models, such as DCRNN, which considers a graph structure.

Next, we combine GCMem with various decoder architectures to investigate how the performance of GCMem changes with different decoder combinations. The results indicate that all decoder combinations achieve state-of-the-art performance for both NAVER-Seoul and METR-LA datasets. This result implies that GCMem effectively learns representations, even with a

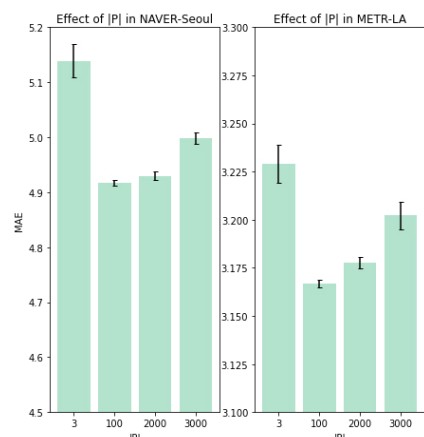

Figure 4: Effect of $|\mathbb{P}|$ in NAVER-Seoul (left) and METR-LA (right).

simple decoder architecture, such as single layer RNN. Also, from the perspective of error accumulation, we notice that PM-MemNet outperforms the CNN decoder even for the 90-minute prediction. In contrast, the one-layered RNN decoder shows a lower performance than the CNN decoder. However, when comparing the RNN decoder to existing RNN-based models, RNN decoder can predict more precisely for the long-term prediction. These results also show that GCMem is a robust solu-

tion for long-term dependency modeling. Modifying the GCMem layer depth, we discover that PM-MemNet generates sufficiently accurate predictions with a single memory layer (i.e., PM-MemNet w/L=1). Although PM-MemNet needs a three-layered GCMem to achieve the highest accuracy, we can still consider deploying the model with a lightweight version while ensuring its state-of-the-art performance.

Next we analyze how different memory sizes (i.e., different pattern numbers) affect the performance. In this experiment, we set $|\mathbb{P}|$ as 3, 100, 2000, and 3000. Note that we set $|\mathbb{P}|$ as 3 to simulate an extreme case, where PM-MemNet has very rare memory space. Figure 4 shows the experimental results. We can observe that PM-MemNet records the best performance with 100 traffic patterns; attaching large $|\mathbb{P}|$ to a model does not always allow the best forecasting performance. We provide the time consumption of the models in Table 4 in Appendix.

## 5    Limitations, Discussion, and Future Directions

This work is the first attempt to design neural memory networks for traffic forecasting. Although effective, as shown in the evaluation results, there are limitations to this approach. First, we extract traffic patterns to memorize them in advance; however, it is possible that the extracted patterns are redundant, even after strict filtering. As such, there is need to find important patterns and optimize the number of memory slots. For example, a future study could investigate how to learn and extend the key space during the training phase (Kaiser et al., 2017). Second, the learning of embedding matrices in PM-MemNet proceeds only based on referred patterns. Because there are no further losses to optimize memory itself, patterns not referred to are not trained. This training imbalance among memories is of interest, as the model cannot generate meaningful representations from rare patterns. A future study may research not only how such a representation imbalance affects performance, but also design a loss function to reduce the representation gap between rare and frequent events. Third, we use cosine similarity in this work, but it may not be an optimal solution, since it causes mismatching with noisy traffic data. Also, the optimal window size for pattern matching remains to be addressed. A future study may focus on approaches to effectively compute the similarity of traffic patterns. Designing a learnable function for the distance measurement is one possible direction. Finally, we show that the model can effectively forecast traffic data with a small group of patterns. This implies a new research direction for comparing results and learning methods that work with sparse data, such as meta learning and few or zero shot learning (Kaiser et al., 2017).

## 6    Conclusion

In this work, we propose PM-MemNet, a novel traffic forecasting model with a graph convolutional memory architecture, called GCMem. By integrating GCNs and neural memory architectures, PM-MemNet effectively captures both spatial and temporal dependency. By extracting and computing representative traffic patterns, we reduce the data space to a small group of patterns. The experimental results for METR-LA and NAVER-Seoul indicate that PM-MemNet outperforms state-of-the-art models. It proves our hypothesis "accurate traffic forecasting can be achieved with a small set of representative patterns" is reasonable. We also demonstrate that PM-MemNet quickly responds to abruptly changing traffic patterns and achieves higher accuracy compared to the other models. Lastly, we discuss the limitations of this work and future research directions with the application of neural memory networks and a small set of patterns for traffic forecasting. In future work, we plan to conduct further experiments using PM-MemNet for different spatio-temporal domains and datasets to investigate whether the insights and lessons from this work can be generalized to other domains.

### Acknowledgements

This work was supported by the Korean National Research Foundation (NRF) grant (No. 2021R1A2C1004542) and by the Institute of Information & Communications Technology Planning & Evaluation (IITP) grants (No. 2020-0-01336–Artificial Intelligence Graduate School Program (UNIST), No. 2021-0-01198–ICT R&D Innovation Voucher Program), funded by the Korea government (MSIT). This work was also partly supported by NAVER Corp.

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

## A  APPENDIX

### A.1  DETAILED EXPERIMENTAL SETUP

**MLP** Multi-layer perceptron with two hidden layers; each layer contains 64 units and rectified linear unit (ReLU) activation.

**STGCN** Spatio-Temporal Graph Convolutional Networks (Yu et al., 2018). STGCN models spatial features using spectral-based graph convolution with Chebyshev polynomial approximation and temporal features using gated CNNs. In our experiment, like the original paper, we set STGCN with a graph convolution kernel size $K = 3$, temporal convolution kernel size of $K_t = 3$, and three layers with 64, 16, and 64 hidden units, respectively.

**DCRNN** Diffusion Convolutional Recurrent Neural Networks (Li et al., 2018). DCRNN is the model that integrates diffusion convolution with RNNs, especially the GRU. In our experiment, both encoder and decoder contain two recurrent layers with 64 hidden units and maximum steps of random walks set as 3 (i.e., $K = 3$).

**GCRNN** Graph Convolutional Recurrent Neural Networks. GCRNN is a variant of DCRNN that integrates simple bidirectional graph convolution and sequence-to-sequence architecture. Both encoder and decoder contain two recurrent layers with 64 hidden units and maximum steps of random walks set as 3 (i.e., $K = 3$).

**ASTGCN** Attention based Spatio-Temporal Graph Convolutional Networks (Guo et al., 2019). ASTGCN models spatio-temporal features by attention, GCNs, and CNNs. For the GCNs, ASTGCN utilizes spectral-based graph convolution with Chebyshev polynomials $K = 3$ and kernel size of CNNs $K_t = 3$. Furthermore, all layers have the same hidden unit size of 64.

**GWNet** Graph WaveNet (Wu et al., 2019). GWNet combines graph convolution and dilated causal convolution. In our experiment, to cover the input sequence length, we utilize 12 layers with dilation factors of $1, 2, 1, 2, \ldots, 1, 2$. Also, following original paper, we set the diffusion step $K = 3$ and $E_1, E_2 \in \mathbb{R}^{N \times 10}$.

**GMAN** Graph Multi-Attention Network (Zheng et al., 2020). GMAN handles both spatial and temporal features using attention and pretrained node embedding. In our experiment, we utilize the same setting as the original paper, that is, three layers for both encoder and decoder with eight attention heads and 64 hidden units (8 hidden units per attention head).

**PM-MemNet** Pattern Matching Memory Networks. Both encoder and decoder have three GCMem layers with 128 hidden units and $|\mathbb{P}| \approx 100$. Note that the decoder also has one GRU cell before GCMem with 128 hidden units. For the $k$-NN, we utilize $k = 3$ and cosine similarity. Also, for the graph convolution, we set the diffusion step as $K = 2$.

In the experiment, we utilize the MAE loss function and Adam optimizer with a learning rate of $1e^{-3}$ for training. The learning rate reduces to $\frac{1}{10}$ for every 10 epochs, starting at 30 epochs, until it reaches $1e^{-6}$. For each model, we trained 100 epochs, with an early termination by monitoring the validation error.

A.2   DETAILED RESULTS WITH 18-STEP INPUT AND OUTPUT SEQUENCES

Table 3: Detailed experimental results with state-of-the-art models with 18-step input and prediction sequences.

| Dataset | T | Metric | HA | MLP | STGCN | GCRNN | DCRNN | ASTGCN | GWNet | GMAN | PM-MemNet |
|---------|---|--------|----|-----|-------|-------|-------|--------|-------|------|-----------|
| NAVER-Seoul | 15min | MAE | 6.54 | 5.28 | 4.63 | 4.87 | 4.86 | 5.09 | 4.91 | 5.20 | **4.57** |
| | | MAPE | 18.24 | 16.86 | 14.49 | 15.23 | 15.35 | 16.14 | 14.86 | 16.98 | **14.43** |
| | | RMSE | 9.32 | 7.78 | 6.92 | 7.18 | 7.12 | 7.44 | 7.24 | 8.32 | **6.72** |
| | 30min | MAE | 7.16 | 6.13 | 5.50 | 5.73 | 5.67 | 5.71 | 5.26 | 5.35 | **5.04** |
| | | MAPE | 20.15 | 20.05 | 17.37 | 18.17 | 18.38 | 18.78 | **16.16** | 17.47 | 16.34 |
| | | RMSE | 10.18 | 9.51 | 8.83 | 9.03 | 8.80 | 8.73 | 8.13 | 8.67 | **7.86** |
| | 45min | MAE | 7.70 | 6.68 | 6.16 | 6.24 | 6.12 | 6.01 | 5.43 | 5.43 | **5.18** |
| | | MAPE | 21.81 | 22.01 | 19.15 | 19.85 | 20.06 | 19.73 | **16.70** | 17.72 | 16.81 |
| | | RMSE | 10.89 | 10.48 | 9.95 | 9.99 | 9.61 | 9.28 | 8.53 | 8.84 | **8.24** |
| | 60min | MAE | 8.22 | 7.08 | 6.77 | 6.58 | 6.40 | 6.22 | 5.55 | 5.48 | **5.24** |
| | | MAPE | 23.37 | 23.44 | 20.42 | 20.95 | 21.09 | 20.37 | 16.97 | 17.89 | **16.94** |
| | | RMSE | 11.54 | 11.13 | 10.89 | 10.58 | 10.06 | 9.58 | 8.77 | 8.94 | **8.39** |
| | 75min | MAE | 8.73 | 7.44 | 7.40 | 6.87 | 6.63 | 6.46 | 5.68 | 5.53 | **5.30** |
| | | MAPE | 24.91 | 24.76 | 21.62 | 21.91 | 21.93 | 20.93 | 17.31 | 18.05 | **17.10** |
| | | RMSE | 12.17 | 11.67 | 11.85 | 11.02 | 10.39 | 9.91 | 9.01 | 9.03 | **8.50** |
| | 90min | MAE | 9.24 | 7.79 | 8.06 | 7.14 | 6.86 | 6.76 | 5.87 | 5.58 | **5.40** |
| | | MAPE | 26.40 | 26.08 | 22.93 | 22.86 | 22.74 | 21.83 | 17.89 | 18.18 | **17.44** |
| | | RMSE | 12.77 | 12.17 | 12.86 | 11.43 | 10.69 | 10.32 | 9.33 | 9.09 | **8.68** |
| METR-LA | 15min | MAE | 4.23 | 2.93 | 2.61 | 2.59 | **2.56** | 3.25 | 2.72 | 2.86 | 2.66 |
| | | MAPE | 9.76 | 7.76 | **6.59** | 6.73 | 6.67 | 9.27 | 7.14 | 7.67 | 7.06 |
| | | RMSE | 7.46 | 5.81 | 5.19 | 5.10 | 6.28 | **5.12** | 5.20 | 5.77 | 5.28 |
| | 30min | MAE | 4.80 | 3.60 | 3.22 | 3.08 | **3.01** | 3.80 | 3.12 | 3.14 | 3.02 |
| | | MAPE | 11.30 | 10.00 | **8.39** | 8.72 | 8.42 | 11.28 | 8.66 | 8.79 | 8.49 |
| | | RMSE | 8.34 | 7.29 | 6.63 | 6.32 | 6.29 | 7.59 | 6.34 | 6.54 | **6.28** |
| | 45min | MAE | 5.32 | 4.17 | 3.78 | 3.46 | 3.34 | 4.20 | 3.39 | 3.34 | **3.24** |
| | | MAPE | 12.71 | 11.93 | 9.84 | 10.26 | 9.71 | 12.64 | 9.63 | 9.55 | **9.33** |
| | | RMSE | 9.18 | 8.37 | 7.76 | 7.12 | 7.08 | 8.40 | 7.06 | 7.00 | **6.87** |
| | 60min | MAE | 5.80 | 4.69 | 4.31 | 3.74 | 3.60 | 4.49 | 3.58 | 3.48 | **3.40** |
| | | MAPE | 14.04 | 13.68 | 11.13 | 11.50 | 10.73 | 13.69 | 10.30 | 10.10 | **9.88** |
| | | RMSE | 9.86 | 9.24 | 8.71 | 7.71 | 7.65 | 8.94 | 7.53 | 7.30 | **7.24** |
| | 75min | MAE | 6.25 | 5.17 | 4.86 | 4.01 | 3.84 | 4.73 | 3.73 | 3.60 | **3.53** |
| | | MAPE | 15.26 | 15.47 | 12.45 | 12.59 | 11.68 | 14.60 | 10.90 | 10.56 | **10.31** |
| | | RMSE | 10.46 | 9.95 | 9.62 | 8.21 | 8.15 | 9.34 | 7.87 | 7.52 | **7.51** |
| | 90min | MAE | 6.65 | 5.58 | 5.41 | 4.23 | 4.06 | 4.97 | 3.85 | 3.71 | **3.64** |
| | | MAPE | 16.37 | 17.08 | 13.76 | 13.49 | 12.53 | 15.53 | 11.39 | 11.00 | **10.74** |
| | | RMSE | 10.97 | 10.52 | 10.47 | 8.79 | 8.58 | 9.71 | 8.12 | 7.71 | **7.74** |

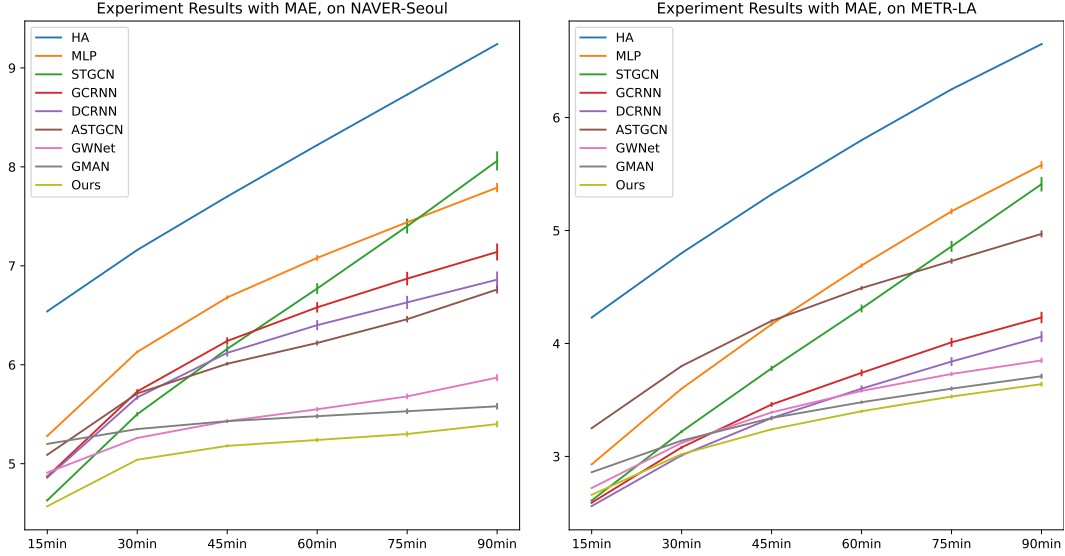

Figure 5: Performance result plots with error bars with NAVER-Seoul (left) and METR-LA (right) datasets.

Table 4: The computation times for each model with METR-LA

| Computation Time | DCRNN | GWNet | GMAN | PM-MemNet | SimpleMem | CNN Decoder | RNN Decoder | PM-MemNet w/ L = 1 |
|---|---|---|---|---|---|---|---|---|
| Training time per epoch (sec) | 691.32 | 102.06 | 500.19 | 196.6 | 169.6 | 104.78 | 39.94 | 127.53 |
| Inference Time (sec) | 56.30 | 6.00 | 9.34 | 14.9 | 12.14 | 10.2 | 4.26 | 10.77 |

## A.3 Detailed Ablation Study Results

Table 5: Detailed ablation study result.

| Dataset | T | Metric | Ours | SimpleMem | CNN Decoder | RNN Decoder | Ours (L=1) | Ours ($\mathbb{P} = k$) | Ours ($\mathbb{P} \gg 1000$) |
|---|---|---|---|---|---|---|---|---|---|
| NAVER-Seoul | 15min | MAE | 4.57 | 5.72 | **4.56** | 4.67 | 4.72 | 4.66 | 4.59 |
| | | MAPE | 14.43 | 18.18 | **14.40** | 14.84 | 14.98 | 14.77 | 14.48 |
| | | RMSE | 6.72 | 8.79 | **6.71** | 6.83 | 6.87 | 6.89 | 6.73 |
| | 30min | MAE | **5.04** | 5.84 | 5.06 | 5.19 | 5.22 | 5.21 | 5.09 |
| | | MAPE | **16.34** | 18.86 | 16.36 | 16.87 | 16.97 | 16.91 | 16.41 |
| | | RMSE | **7.86** | 9.24 | 7.90 | 8.02 | 8.03 | 8.18 | 7.97 |
| | 45min | MAE | **5.18** | 6.10 | 5.23 | 5.38 | 5.43 | 5.43 | 5.27 |
| | | MAPE | **16.81** | 19.98 | 16.98 | 17.61 | 17.66 | 17.72 | 16.97 |
| | | RMSE | **8.24** | 9.71 | 8.32 | 8.47 | 8.49 | 8.69 | 8.45 |
| | 60min | MAE | **5.24** | 6.38 | 5.32 | 5.47 | 5.52 | 5.53 | 5.36 |
| | | MAPE | **16.94** | 21.42 | 17.19 | 17.91 | 17.97 | 18.05 | 17.19 |
| | | RMSE | **8.39** | 10.08 | 8.51 | 8.69 | 8.70 | 8.92 | 8.67 |
| | 75min | MAE | **5.30** | 6.65 | 5.39 | 5.56 | 5.60 | 5.62 | 5.42 |
| | | MAPE | **17.10** | 22.52 | 17.38 | 18.20 | 18.23 | 18.32 | 17.40 |
| | | RMSE | **8.50** | 10.48 | 8.62 | 8.86 | 8.85 | 9.09 | 8.80 |
| | 90min | MAE | **5.40** | 6.95 | 5.55 | 5.70 | 5.72 | 5.74 | 5.52 |
| | | MAPE | **17.44** | 23.89 | 17.99 | 18.73 | 18.63 | 18.73 | 17.76 |
| | | RMSE | **8.68** | 10.88 | 8.82 | 9.10 | 9.05 | 9.31 | 8.94 |
| METR-LA | 15min | MAE | 2.66 | 3.01 | **2.63** | 2.68 | 2.68 | 2.67 | 2.68 |
| | | MAPE | 7.06 | 8.03 | **6.98** | 7.10 | 7.11 | 7.09 | 7.13 |
| | | RMSE | **5.28** | 5.94 | 5.32 | 5.31 | 5.31 | 5.35 | 5.31 |
| | 30min | MAE | 3.02 | 3.27 | **3.01** | 3.06 | 3.06 | 3.06 | 3.04 |
| | | MAPE | 8.49 | 9.20 | **8.46** | 8.56 | 8.59 | 8.67 | 8.51 |
| | | RMSE | 6.28 | 6.68 | 6.36 | 6.32 | **6.27** | 6.36 | 6.32 |
| | 45min | MAE | **3.24** | 3.52 | 3.25 | 3.30 | 3.30 | 3.32 | 3.28 |
| | | MAPE | **9.33** | 10.18 | 9.30 | 9.44 | 9.47 | 9.67 | 9.31 |
| | | RMSE | 6.87 | 7.28 | 6.94 | 6.93 | **6.86** | 7.00 | 6.93 |
| | 60min | MAE | **3.40** | 3.72 | 3.41 | 3.46 | 3.47 | 3.49 | 3.45 |
| | | MAPE | **9.88** | 10.94 | 9.88 | 10.02 | 10.07 | 10.34 | 9.87 |
| | | RMSE | **7.24** | 7.70 | 7.28 | 7.31 | 7.25 | 7.39 | 7.32 |
| | 75min | MAE | 3.53 | 3.91 | **3.52** | 3.59 | 3.60 | 3.62 | 3.58 |
| | | MAPE | **10.31** | 11.61 | 10.35 | 10.47 | 10.56 | 10.85 | 10.27 |
| | | RMSE | 7.51 | 8.06 | **7.50** | 7.59 | 7.53 | 7.67 | 7.59 |
| | 90min | MAE | **3.64** | 4.09 | 3.65 | 3.71 | 3.73 | 3.75 | 3.69 |
| | | MAPE | **10.74** | 12.25 | 10.85 | 10.87 | 10.98 | 11.30 | 10.63 |
| | | RMSE | 7.74 | 8.38 | **7.71** | 7.81 | 7.75 | 7.91 | 7.81 |

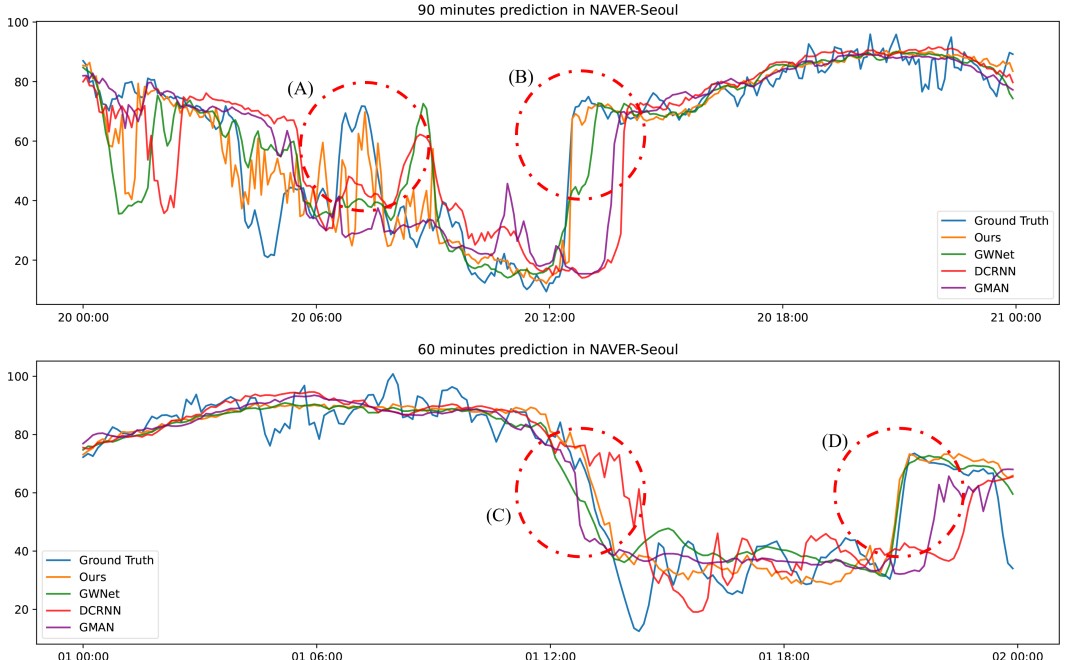

Figure 6: NAVER-Seoul speed prediction visualization for (upper) 90-minute forecasting and (lower) 60-minute forecasting. Red circles show the interval with abrupt speed changing, which are successfully detected by PM-MemNet.

### A.4 QUALITATIVE EVALUATION

We present a qualitative analysis using the NAVER-Seoul dataset, which contains a complex road network and the dynamic traffic conditions of urban areas. In this analysis, we evaluate whether our approach effectively patterned traffic data for the traffic forecasting problem. If the approach is valid, we expect PM-MemNet to be more accurate in predicting difficult trailing patterns due to the high correlation between leading and trailing patterns. To verify our expectations, we visualize the long-term traffic prediction results, as shown in Figure 6 (top), where PM-MemNet predicts the end of congestion at 12:00 PM (B) more accurately. Also, in 7:00 AM (A), PM-MemNet quickly catches up on the unexpected peak speed (this continues for about 30 minutes, and PM-MemNet efficiently catches up within 15 minutes). From Figure 6 (bottom), we can see that PM-MemNet predicts a speed drop and its recovery more accurately compared to the other models. For example, GMAN and GWNet predict the speed drop earlier than the actual speed drop, and DCRNN predicts the speed drop later than the real speed drop, as shown in 6 (C). However, PM-MemNet predicts the occurrence of slowdowns on time, even for long-term traffic prediction (D). Overall, with representative traffic patterns and memorization, PM-MemNet effectively handles abruptly changing traffic conditions, even for long-term predictions.

Next, we analyze how PM-MemNet effectively matches input patterns with memorized patterns. Figure 7 presents three examples, each of which has two time-series lines. The red line denotes the current time step $t$. Before the current time step $t$, The blue and orange line means input and corresponding matched pattern, respectively. After the current time step $t$, The blue and orange line means ground truth and prediction results, respectively. In each example, we find that the model effectively retrieves a memorized pattern that matched the input sequence well for prediction. Note that all examples show how PM-MemNet operates for difficult roads and time with rapid speed drops.

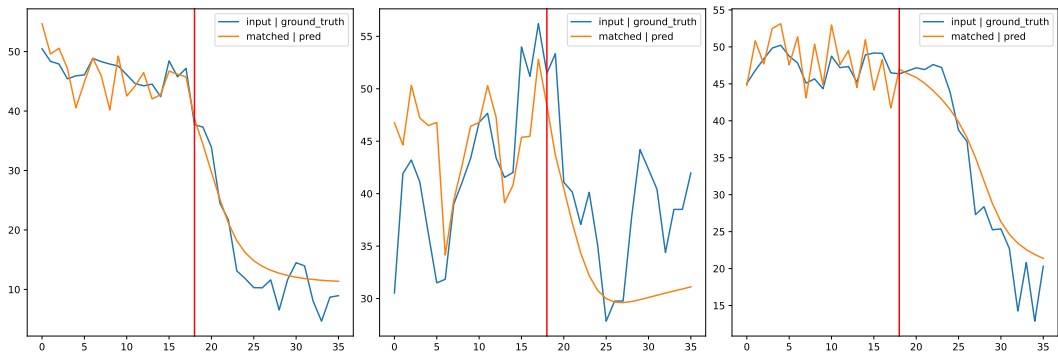

Figure 7: Sample input (ground truth, blue) and matched representative pattern for prediction (orange). the red line means the current time step $t$.

## A.5 EXPERIMENT WITH 12 SEQUENCE SETTING

Table 6: Experimental results with state-of-the-art models on common 12 sequence prediction setting. For PEMS-BAY, $\delta = 0.9$ and $|\mathbb{P}| \approx 100$.

| Dataset | T | Metric | HA | MLP | STGCN | GCRNN | DCRNN | ASTGCN | GWNet | GMAN | PM-MemNet |
|---|---|---|---|---|---|---|---|---|---|---|---|
| NAVER-Seoul | 15min | MAE | 6.22 | 5.28 | 4.69 | 5.00 | 4.92 | 4.91 | **4.50** | 4.90 | 4.52 |
| | | MAPE | 19.68 | 16.69 | 14.54 | 15.75 | 15.54 | 15.55 | 14.42 | 15.72 | **14.27** |
| | | RMSE | 9.54 | 7.78 | 7.02 | 7.34 | 7.20 | 7.23 | **6.61** | 7.64 | 6.67 |
| | 30min | MAE | 6.86 | 6.14 | 5.60 | 5.86 | 5.76 | 5.37 | 5.05 | 5.14 | **5.01** |
| | | MAPE | 21.77 | 19.92 | 17.61 | 18.86 | 18.59 | 17.14 | 16.71 | 16.61 | **16.10** |
| | | RMSE | 10.66 | 9.53 | 8.99 | 9.15 | 8.92 | 8.31 | **7.82** | 8.24 | 7.82 |
| | 60min | MAE | 7.90 | 7.10 | 6.89 | 6.75 | 6.55 | 5.86 | 5.44 | 5.38 | **5.30** |
| | | MAPE | 25.05 | 23.44 | 20.77 | 21.97 | 21.53 | 18.42 | 17.84 | 17.51 | **17.03** |
| | | RMSE | 12.28 | 11.16 | 11.01 | 10.75 | 10.29 | 9.16 | 8.57 | 8.71 | **8.51** |
| | Avg. | MAE | 6.87 | 5.82 | 5.39 | 5.55 | 5.43 | 5.16 | 4.74 | 5.10 | **4.69** |
| | | MAPE | 21.77 | 18.84 | 16.67 | 17.76 | 17.49 | 16.79 | 15.44 | 16.48 | **14.97** |
| | | RMSE | 10.70 | 8.96 | 8.52 | 8.60 | 8.34 | 8.21 | **7.28** | 8.12 | 7.28 |
| METR-LA | 15min | MAE | 3.75 | 2.92 | 2.88 | 2.80 | 2.73 | 3.07 | 2.69 | 2.81 | **2.65** |
| | | MAPE | 10.01 | 7.71 | 7.62 | 7.50 | 7.12 | 5.90 | **6.98** | 7.43 | 7.01 |
| | | RMSE | 7.31 | 5.82 | 5.74 | 5.51 | 5.27 | 8.23 | **5.16** | 5.55 | 5.29 |
| | 30min | MAE | 4.37 | 3.60 | 3.47 | 3.24 | 3.13 | 3.61 | 3.08 | 3.12 | **3.03** |
| | | MAPE | 11.85 | 9.93 | 9.57 | 9.00 | 8.65 | 10.34 | 8.43 | **8.35** | 8.42 |
| | | RMSE | 8.52 | 7.31 | 7.24 | 6.74 | 6.40 | 7.16 | **6.21** | 6.46 | 6.29 |
| | 60min | MAE | 5.45 | 4.70 | 4.59 | 3.81 | 3.58 | 4.42 | 3.53 | **3.46** | 3.46 |
| | | MAPE | 15.04 | 13.64 | 12.70 | 10.90 | 10.43 | 13.35 | 10.05 | 10.06 | **9.97** |
| | | RMSE | 10.39 | 9.26 | 9.40 | 8.16 | 7.60 | 8.73 | 7.31 | 7.37 | **7.29** |
| | Avg. | MAE | 4.43 | 3.63 | 3.64 | 3.28 | 3.14 | 3.61 | 3.09 | 3.13 | **2.99** |
| | | MAPE | 12.02 | 10.07 | 9.96 | 9.13 | 8.72 | 10.32 | 8.42 | 8.61 | **8.27** |
| | | RMSE | 8.66 | 7.23 | 7.46 | 6.80 | 6.42 | 7.18 | 6.26 | 6.46 | **6.14** |
| PEMS-BAY | 15min | MAE | 2.26 | 1.49 | 1.42 | 1.34 | 1.33 | 1.55 | 1.30 | 1.36 | **1.27** |
| | | MAPE | 5.03 | 3.15 | 3.10 | 2.79 | 2.78 | 3.44 | **2.73** | 2.93 | 2.75 |
| | | RMSE | 5.18 | 3.24 | 3.08 | 2.82 | 2.79 | 3.17 | **2.74** | 2.88 | 2.80 |
| | 30min | MAE | 2.72 | 2.03 | 1.91 | 1.72 | 1.68 | 2.01 | 1.63 | 1.64 | **1.62** |
| | | MAPE | 6.18 | 4.55 | 4.49 | 3.87 | 3.78 | 4.66 | 3.67 | 3.71 | **3.66** |
| | | RMSE | 6.26 | 4.70 | 4.44 | 3.85 | 3.77 | 4.19 | 3.70 | 3.78 | **3.65** |
| | 60min | MAE | 3.52 | 2.80 | 2.53 | 2.08 | 2.01 | 2.57 | 1.95 | **1.90** | 1.95 |
| | | MAPE | 8.19 | 6.86 | 6.23 | 5.01 | 4.76 | 6.01 | 4.63 | **4.45** | 4.64 |
| | | RMSE | 7.94 | 6.34 | 5.92 | 4.72 | 4.59 | 5.27 | 4.52 | **4.40** | 4.51 |
| | Avg. | MAE | 2.76 | 2.02 | 1.88 | 1.66 | 1.62 | 1.97 | 1.57 | 1.58 | **1.55** |
| | | MAPE | 6.29 | 4.61 | 4.28 | 3.75 | 3.64 | 4.54 | 3.67 | **3.59** | 3.62 |
| | | RMSE | 6.40 | 4.57 | 4.30 | 3.66 | 3.58 | 4.18 | 3.47 | 3.58 | **3.44** |

