# OpenReview forum: "Learning to Remember Patterns: Pattern Matching Memory Networks for Traffic Forecasting"
_ICLR.cc/2022/Conference — ICLR 2022 Poster_

### Official Review · Reviewer_mucH · 2021-11-01

**Correctness:** 3
**Technical Novelty And Significance:** 3
**Empirical Novelty And Significance:** 2
**Recommendation:** 5
**Confidence:** 4

**Main Review:**

This paper studies the traffic forecasting problem and proposes to conduct prediction by pattern matching. Authors first extract key patterns from the historical data in an offline manner and then fetch the patterns for each time series with a distance function (e.g., cosine similarity). Then, the patterns of different nodes are interacted with GCN to get node representation.
While using pattern matching for traffic forecasting is reasonable and the design contains some new ideas, the paper should be carefully improved on the following points:
1. The writing is not clear. Especially, the descriptions of the extracted patterns are not enough, are they just averaged speed? Are they representative? How to sample the patterns with a time window? Eq.1 should be a sum from j=0 to k? What is $p_1$ to get the noise in Eq.4?
2. In section 3.3, there lacks justifications about why capturing interactions among fetched patterns is necessary and also why pattern level attention is necessary?
3. The experimental results are wired. I doubt the necessity of using a different setting (e.g., 18 steps instead of 12) with existing works, which makes it hard to conduct a fair comparison. And the experimental results of some baseline methods are lower than the reported performance. I would strongly suggest authors use the same settings with existing works to demonstrate their performance and also take PeMS (e.g., PeMS-bay) dataset into consideration.
4. Many traffic forecasting works use MLP for prediction directly, does the proposed decoder performs better than this baseline?
5. There are many grammar mistakes in the paper, such as:
	* “The goal in this work is different the previous work”
	* “Similarity distribution of original pattern” in Figure 2
	* “we further use a adjacency matrix”


**Summary Of The Paper:**

This paper studies the traffic forecasting problem and proposes to conduct prediction by pattern matching. Authors first extract key patterns from the historical data in an offline manner and then fetch the patterns for each time series with a distance function (e.g., cosine similarity). Then, the patterns of different nodes are interacted with GCN to get node representation.

**Summary Of The Review:**

1. The writing of the paper should be carefully improved;
2. Current experimental results can not demonstrate the effectiveness of the method;

---

> ### Author Response · Authors · 2021-11-20
> **Respond to Reviewer mucH - Part 2**
>
> >> “Are they representative? “
>
>
> Yes, we believe that the extracted patterns sufficiently represent the datasets, as indicated by the performance results. We extracted the patterns from real-world traffic data and performed cluster-based undersampling to carefully select representative traffic patterns. On the other hand, we agree there could be different ways to extract representative patterns. A future study can investigate how to extract more representative patterns to improve the model’s performance further.
>
> >> “How to sample the patterns with a time window?”
>
>
> We collected the daily patterns from roads and sliced them with the given window size T’. The divided time-series data are what we call patterns for memory networks (See Figure 2). Further, before we train our model, we noted an imbalance in the patterns, so we undersampled the pattern set P by taking cluster centers formed by our distance function d and the threshold \delta. Note that p and p’ are in the same cluster if d(p, p’) > \delta.
> We clarify this in the paper.
>
> >> “Eq.1 should be a sum from j=0 to k?”
>
>
> >> “What is p1 to get the noise in Eq.4?”
>
>
> No. Eq.1 is the sum from j = 1 to k. We clarify this point in the paper.
> As we have indicated in Section 3.1, p_1 in the Eq.4 is the first nearest neighboring pattern of X_i.
>
>
> >> “In section 3.3, there lacks justifications about why capturing interactions among fetched patterns is necessary and also why pattern level attention is necessary?”
>
>
> We used the pattern-level attention to connect the memory context of layer l and the memory context of the next layer {l+1}. This technique, called “adjacent weight tying,” has been widely used in prior studies, including in the work done by Madotto et al. (2018), leading to performance improvement. We clarified the role and benefit of the technique further in Section 3.3.
>
>
> >> “I doubt the necessity of using a different setting (e.g., 18 steps instead of 12) with existing works, which makes it hard to conduct a fair comparison.”
>
>
> We conducted an additional experiment with 12 steps and found that our model shows better performance compared to most state-of-the-art models. Considering that this work is the first attempt at using memory networks and experiment results, we believe there is a high chance for memory network models to significantly outperform previous models in the future.
> We added the experiment results to Appendix (Table 6).
>
> >> “And the experimental results of some baseline methods are lower than the reported performance. I would strongly suggest authors use the same settings with existing works to demonstrate their performance”
>
>
> As we stated in Section 4.1, we used the source code and experiment settings provided by the authors. To clarify further, we provided the hyperparameter settings that we used for the models in the experiment in Appendix A.1. We conjecture that the performance difference is attributable to the different input steps, so we conducted another experiment with 12 steps. Notably, we found that our model still shows better performance.
>
>
> >> “also take PeMS (e.g., PeMS-bay) dataset into consideration.”
>
>
> We conduct an additional experiment with PeMS-bay and find that our model shows better performance on average compared to most state-of-the-art models. Considering that this work is the first attempt at using memory networks and experiment results, we believe there is a high chance for memory network models to significantly outperform previous models in the future.
>
>
> >> “Many traffic forecasting works use MLP for prediction directly, does the proposed decoder perform better than this baseline?”
>
>
> We conducted experiments with MLP and updated the result in Table 1, Table 3, and Table 6.
>
>
> >> “There are many grammar mistakes in the paper”
>
>
> Answer. We fixed all grammar errors. We also work with native English speakers to further improve the paper writing.

---

> ### Author Response · Authors · 2021-11-20
> **Respond to Reviewer mucH - Part 1**
>
> >> “The writing is not clear. Especially, the descriptions of the extracted patterns are not enough, are they just averaged speed?“
>
>
> Many studies have revealed that traffic patterns are not that different at similar time ranges in days (i.e., temporal dependency) [1-3]. Also, it has re-occurring patterns at particular time ranges, such as during rush-hour traffic congestion [1-3]. Therefore, we first computed the daily traffic pattern of a road by averaging 5-minute interval historical speed data of 24 hours. We then divided the daily patterns with the given window size T’. These are what we call traffic patterns in P.
> We revised Section 3.2 to clarify this point.
>
> [1] Ali Zonoozi, Jung-jae Kim, Xiao-Li Li, and Gao Cong. Periodic-crn: A convolutional recurrent model for crowd density prediction with recurring periodic patterns. IJCAI, pp. 3732–3738, 2018.
> [2] Muhua Zheng, Zhongyuan Ruan, Ming Tang, Younghae Do, and Zonghua Liu. Influence of periodic traffic congestion on epidemic spreading. International Journal of Modern Physics C, 27(05): 1650048, 2016
> [3] Wanida Liyong and Peerapon Vateekul. Improve traffic prediction using accident embedding on ensemble deep neural networks. KST, pp. 11–16, 2019

---

### Official Review · Reviewer_xDZJ · 2021-11-02

**Correctness:** 2
**Technical Novelty And Significance:** 2
**Empirical Novelty And Significance:** 2
**Recommendation:** 6
**Confidence:** 4

**Main Review:**

Strengths:
- The idea of using both extracted traffic patterns and spatial dependencies is interesting.

Weakness:

One of the major concerns is that the paper is not well presented, and critical information on the model and problem setup is missing.
- In the problem setup, the definition of traffic patterns is not clear. Based on Figure 1, it seems to be a sequence of traffic speeds along the timeline.
- Section 3.2 is difficult to follow. It says that traffic patterns are average speed on each road. Intuitively, the traffic situation of each road - could be very different. Sometimes, even on the same road, the traffic situation in different directions could be very different.
- The author's claim that the pattern set is imbalanced is unconvincing since the only evidence provided is the one arbitrary pattern in Figure 2.
- Cosine similarity is a continuous value. However, the authors treat them as discrete bins without mentioning the motivation.
- The input, output of each component is unclear. In particular, the connection between each component is not clearly described in either the overview figure or the model section.  For example, it's unknown how the p_j memory context m_j is obtained. In equation 1, M_j^l's calculation is independent with layer l, which means M_j^l, m_j^{l + 1}, ... etc., all have the same value. It doesn't make sense. If C_{i,j}^l = \alpha_{i,j}^l, then why use this new symbol C_{i,j}^l. The dimension of node embedding E_1 and E_2 is not clear. The encoder and decoder structure is unclear, either.

Another concern is that the paper's contribution seems incremental. Just as the authors stated, GCMeme is a combination of (Madotto et al., 2018) and (Bruna et al., 2014), which takes care of memory attention and graph convolution separately.

The last concern is that some important baselines, such as ASTGCN, STGCN, etc., are missing. Since table 1 includes the main result, it is necessary to include the error bars instead of simply providing the prediction results.

**Summary Of The Paper:**

This work studied the problem of traffic speed forecasting. In particular, the authors proposed a framework that improves forecasting performance by leveraging both the spatio-temporal dependency and extracted traffic patterns.

**Summary Of The Review:**

The paper is not well presented, and critical information regarding the problem and model setup is missing, which makes the paper difficult to understand.

---

> ### Author Response · Authors · 2021-11-20
> **Respond to xDZJ - Part 2**
>
>
> >> “Cosine similarity is a continuous value. However, the authors treat them as discrete bins without mentioning the motivation.”
> To fix this, we replaced the existing plot with density plots, as shown in Figure 2 (b, c).
> >> “The input, output of each component is unclear. In particular, the connection between each component is not clearly described in either the overview figure or the model section. For example, it's unknown how the p_j memory context m_j is obtained. In equation 1, M_j^l's calculation is independent with layer l, which means M_j^l, m_j^{l + 1}, ... etc., all have the same value. It doesn't make sense. If C_{i,j}^l = \alpha_{i,j}^l, then why use this new symbol C_{i,j}^l. The dimension of node embedding E_1 and E_2 is not clear. The encoder and decoder structure is unclear, either.”
>
>
>  We revised Section 3.3, 3.4, and 3.5 for clarification.
> The input and output of each component were updated. Also, to help the reader’s understanding, we modified Figure 3. Now, reader can refer to Figure 3 (a), Section 3.4, and 3.5 to check the encoder and decoder architecture.
> In Section 3.4. we described where the final hidden state h^L is used.
> In Section 3.5, we updated the information about the GRU hidden state.
>
>
>
> >> “it's unknown how the p_j memory context m_j is obtained. In equation 1, M_j^l's calculation is independent with layer l, which means M_j^l, m_j^{l + 1}, ... etc., all have the same value. It doesn't make sense.”
>
>
> → We fixed the equation and existing errors in Section 3.3. We further clarified below:
> m_j is not the same in layers. We fixed it (m_j → m_j^l)
> Each layer has its own embedding matrix M_l for memorization
> For each p_j, m^l_j is the corresponding learnable embedding vector of l-th layer.
>
> >> “If C_{i,j}^l = \alpha_{i,j}^l, then why use this new symbol C_{i,j}^l”
>
> → By the equation, we mean \alpha_{i,j}^l is the entry in i-th row and j-th column of the matrix C^l. We clarify this in the paper.
>
> >> “The dimension of node embedding E_1 and E_2 is not clear”
>
> → We double-checked all parameters and notations and added dimensions.
>  e.g., E_1 \in \mathbb{R}^{N \times d_h}
>
> >> “Another concern is that the paper's contribution seems incremental. Just as the authors stated, GCMeme is a combination of (Madotto et al., 2018) and (Bruna et al., 2014), which takes care of memory attention and graph convolution separately.”
>
> As we articulated in the Introduction, this work has multiple contributions, one of which is the implementation of GCMem. We argue that the contribution of this work should be considered in a more holistic way. We believe that the primary contribution of this work and the results obtained is the introduction of a new approach in the sense that we hypothesized and demonstrated that traffic forecasting can be effectively performed with a small set of extracted and memorized representative traffic patterns. We conjecture that the newly introduced method would inspire many future studies in different domains, such as effective methods for representative traffic pattern extraction and memorization, end-to-end methods from pattern extraction to prediction, and novel model designs for better utilization of memory networks.
> In terms of datasets, there have been various public datasets for traffic forecasting, such as METR-LA and PEMS-BAY. Although useful for evaluation, they can be thought of as somewhat easy data with many static highway traffic patterns compared with the traffic patterns of major cities’ internal roads. As such, we evaluated the models by using an additional dataset, namely, NAVER-Seoul (collected in Seoul, Korea), to investigate the performance of existing models and our new deep learning models with complicated traffic patterns. To contribute to the community further, we will open the NAVER-Seoul data with the city's road connections, so that researchers and practitioners in the field can better investigate and evaluate the deep learning models for traffic forecasting.
>
> >> “The last concern is that some important baselines, such as ASTGCN, STGCN, etc., are missing”
>
> → We conducted additional experiments, comparing our model to ASTGCN and STGCN, and added results to the paper (Table 1).
> We further perform an experiment with the 12 sequence input and 12 sequence output sequence setting, a conventional evaluation setting, and find that our model outperforms existing models. Detailed results are in Appendix Table 6.
>
> >> “Since table 1 includes the main result, it is necessary to include the error bars instead of simply providing the prediction results.”
>
> → We provided two charts of error bars in Appendix (Figure 5). We note this in Section 4.1.

---

> ### Author Response · Authors · 2021-11-20
> **Respond to xDZJ - Part 1**
>
> We sincerely thank the reviewer for detailed and helpful comments.
>
> >> “In the problem setup, the definition of traffic patterns is not clear. Based on Figure 1, it seems to be a sequence of traffic speeds along the timeline.”
>
>
> By traffic patterns, we mean the representative traffic speeds (or flows) of the roads, each of which has the same dimensionality of X_i.
> In Section 3.1, we defined set P as a subset of R^{T’}, where element p is an arbitrarily extracted traffic speed that has the same dimensionality as X_i.
>
> >> “Section 3.2 is difficult to follow. It says that traffic patterns are average speed on each road. Intuitively, the traffic situation of each road - could be very different. Sometimes, even on the same road, the traffic situation in different directions could be very different.”
>
>
> We fully revised Section. 3.2 to explain and clarify this issue further.
> We first computed the daily traffic pattern of a road. To compute the daily traffic pattern of a road, we averaged its historical speed data (24 hours, 5-minute interval), as many studies have revealed that traffic patterns are not that different at similar time ranges in days (i.e., temporal dependency) and that very similar patterns re-occur at some time ranges, such as during the morning rush-hour traffic congestion [7–9]. We then divided the daily patterns with the given window size. These are what we call traffic patterns in P.
> Yes, the traffic situations in different directions could be very dissimilar (See Figure 1 in [1,2]). Therefore, traffic datasets distinguish the roads by direction as well (i.e., upstream/downstream).
> As such, existing models that have been trained using the datasets make different predictions based on the road direction and time [2–6]. You can see the term “bi-direction” or “directed” in these studies, which refers to considerations regarding road directions.
>
>
> [1] Yaguang Li and Cyrus Shahabi. A brief overview of machine learning methods for short-term traffic forecasting and future directions. SIGSPATIAL Special, 10(1):3–9, 2018.
>
>
> [2] Yaguang Li, Rose Yu, Cyrus Shahabi, and Yan Liu.  Diffusion convolutional recurrent neural network: Data-driven traffic forecasting. In International Conference on Learning Representations, 2018.
>
>
> [3] Bing Yu, Haoteng Yin, and Zhanxing Zhu.  Spatio-temporal graph convolutional networks: A deep learning framework for traffic forecasting.  In Proceedings of the International Joint Conference on Artificial Intelligence, pp. 3634–3640, 2018.
>
>
> [4] Shengnan Guo, Youfang Lin, Ning Feng, Chao Song, and Huaiyu Wan. Attention based spatial-temporal graph convolutional networks for traffic flow forecasting. In Proceedings of the AAAI Conference on Artificial Intelligence, volume 33, pp. 922–929, 2019.
>
>
> [5] Zonghan Wu, Shirui Pan, Guodong Long, Jing Jiang, and Chengqi Zhang. Graph wavenet for deep spatial-temporal graph modeling.  In Proceedings of the International Joint Conference on Artificial Intelligence, pp. 1907–1913, 2019.
>
>
> [6] Chuanpan Zheng, Xiaoliang Fan, Cheng Wang, and Jianzhong Qi. GMAN: A graph multi-attention network for traffic prediction. In Proceedings of the AAAI Conference on Artificial Intelligence,pp. 1234–1241, 2020
>
>
> [7] Ali Zonoozi, Jung-jae Kim, Xiao-Li Li, and Gao Cong. Periodic-crn: A convolutional recurrent model for crowd density prediction with recurring periodic patterns. In Proceedings of the International Joint Conference on Artificial Intelligence, pp. 3732–3738, 2018.
>
>
> [8] Muhua Zheng, Zhongyuan Ruan, Ming Tang, Younghae Do, and Zonghua Liu. Influence of periodic traffic congestion on epidemic spreading. International Journal of Modern Physics C, 27(05): 1650048, 2016
>
>
> [9] Wanida Liyong and Peerapon Vateekul. Improve traffic prediction using accident embedding on ensemble deep neural networks. In International Conference on Knowledge and Smart Technology, pp. 11–16, 2019
>
>
> >> “The author's claim that the pattern set is imbalanced is unconvincing since the only evidence provided is the one arbitrary pattern in Figure 2.”
>
>
> We provided more explanation and examples on the imbalance in Section 3.2.
> Now, Figure 2 (b) shows a similar distribution across all patterns instead of one arbitrary pattern. Notably, according to similarity distribution, there are too many similar patterns in P.

---

### Official Review · Reviewer_Cncx · 2021-11-02

**Correctness:** 4
**Technical Novelty And Significance:** 4
**Empirical Novelty And Significance:** 4
**Recommendation:** 8
**Confidence:** 5

**Main Review:**

This paper is exciting. It proposes a new way of looking at a very well-studied problem in traffic management. The new approach is novel, technically sound, and well motivated. As a first step, there is much opportunity for further research into more sophisticated models for intelligent pattern extraction, filtering, and inference.

I wholeheartedly recommend acceptance not simply because the idea is novel, but because I strongly believe this paper will birth a new direction of research into traffic flow forecasting.

In order to further improve the paper, I suggest the following:
1. Currently, the results in Table 1 do not look too impressive. This may call the significance of the approach into question. Do you have any opinion on how to further improve performance?

2. Can you provide visual results for several examples where you show an input sequence, its extracted pattern, and finally the resulting prediction and comparison with ground truth.

3. The paper exposition can be improved significantly. For example, in 3.2, what do you mean by "zero-based signals", "without any duplication in range". Such cryptic phrases are scattered throughout the paper. This concern has also been raised by other reviewers, but in my opinion, assuming that the authors improve the clarity, the paper could be accepted.

**Summary Of The Paper:**

The paper proposes a new approach for macroscopic traffic flow forecasting in which the traffic forecasting problem is reformulated as a key-value pair matching problem as opposed to the conventional way of using a neural network to predict a sequence given past input. Results have been provided on two datasets along with ablation results.




**Summary Of The Review:**

I recommend acceptance.

---

> ### Author Response · Authors · 2021-11-20
> **Respond to Reviewer Cncx**
>
> Thank you for your valuable suggestions.
> >> “Currently, the results in Table 1 do not look too impressive. This may call the significance of the approach into question. Do you have any opinion on how to further improve performance?”
>
>
> Many possible approaches and methods can be used to improve the performance further. Some examples include the use of alternative methods for extracting more appropriate representative patterns and designing new models to utilize and manage the memory networks better. We suggest that usage of advanced distance functions for query-key matching, such as the learnable distance function or distance function in the latent domain, could further provide a new opportunity to improve performance. Additionally, meta-learning for memorization can be one solution for better performance. For more detailed information, please refer to Section 5.
>
>
> >> “Can you provide visual results for several examples where you show an input sequence, its extracted pattern, and finally the resulting prediction and comparison with ground truth.”
>
>
> We provided three additional examples in the Appendix (Figure 7). In the examples, the x-axis represents the different time steps (total of 36 steps, 5-minute interval), while the y-axis represents the road speed. The input sequence is the time series before Step 18 (red vertical line). The extracted pattern used for prediction is shown as the time series after Step 18. The blue line represents the ground truth data. For more detailed information and explanation, please refer to Appendix A.4.
>
>
> >> “The paper exposition can be improved significantly. For example, in 3.2, what do you mean by "zero-based signals", "without any duplication in range". Such cryptic phrases are scattered throughout the paper. This concern has also been raised by other reviewers, but in my opinion, assuming that the authors improve the clarity, the paper could be accepted.”
>
>
> We revised the whole paper, including Section 3.2, to enhance clarity. During the revision, the pointed phrases were removed. Also, we fixed other cryptic, not defined words or sentences.

---

> > ### Comment · Reviewer_Cncx · 2021-11-28
> > **Thanks for the response**
> >
> > Thank you. Response is appropriate. All the best !

---

### Official Review · Reviewer_L94p · 2021-11-02

**Correctness:** 4
**Technical Novelty And Significance:** 3
**Empirical Novelty And Significance:** 3
**Recommendation:** 8
**Confidence:** 2

**Main Review:**

Strengths
S1 The idea of using a memory network to remember traffic patterns is very interesting. It has great research value.
S2 The experiment results are promising.
S3 The paper is well written and easy to follow.

Weaknesses
W1 It is interesting to know how the size of P affects the prediction accuracy. An ablation study on it (and other configurations) would potentially make this paper more interesting.



**Summary Of The Paper:**

This paper explores a new direction of model design in traffic forecasting tasks. It proposes a neural memory module to model the spatio-temporal traffic data and designs a new traffic forecasting model based on the memory module. Experiments on a few public datasets demonstrate the effectiveness of the proposed scheme.

**Summary Of The Review:**

This is a good paper - it proposes a novel idea to solve the traffic prediction problem and conducts sufficient experiments to prove the effectiveness of the proposed idea.

---

> ### Author Response · Authors · 2021-11-20
> **Respond to Reviewer L94p**
>
> Thank you for your constructive comment.
> >> "Weaknesses W1 It is interesting to know how the size of P affects the prediction accuracy. An ablation study on it (and other configurations) would potentially make this paper more interesting."
>
> We conducted an additional experiment and presented the results of the ablation study (Table 2 and Figure 4). The results indicate that both too small and too large sizes of memory do not allow the best performance. Please note that we produced the best results on Naver-Seoul and METR-LA when |P| was 100 (i.e., 100 representative patterns) by setting the delta to 0.7 and 0.9, respectively. We refer to the ablation study results in Table 2 and Figure 4.

---

### Author Response · Authors · 2021-11-20
**Revision Summary**

We would like to appreciate the thoughtful and detailed comments by the reviewers.
After revision, we changed and updated below:

 - Updated uncertain words and sentences, especially in Section 3.
 - Because Section 3.2 is uncertain, we fully rewrite Section 3.2
 - We conduct more experiments with additional models (i.e., STGCN and ASTGCN) and settings (i.e., the number of representative patterns)
 - For the experiment setup, we provide all of the detailed experimental settings in Appendix A.1
 - In the paper, we have moved qualitative evaluation to Appendix. Also, in qualitative evaluation, we provide some examples including input, matched pattern, ground truth, and prediction.
 - In Appendix, we updated experimental results with common setting, T = T' = 12. In the common setting, we further conduct experiments for the additional dataset, PEMS-BAY
 - To enhance readability, we updated Figure 2 and Figure 3.

---

### Decision · Program_Chairs · 2022-01-20

**Decision:**

Accept (Poster)

**Comment:**

The paper presents a neural architecture based on neural memory modules to model the spatiotemporal traffic data. The reviewers think this is an important application of deep learning and thus fits the topic of ICLR. The writing and the novelty of the proposed method need improvement.